# High-Throughput Fluorescent Assay for Inhibitor Screening of Proteases from RNA Viruses

**DOI:** 10.3390/molecules26133792

**Published:** 2021-06-22

**Authors:** Bara Cihlova, Andrea Huskova, Jiri Böserle, Radim Nencka, Evzen Boura, Jan Silhan

**Affiliations:** Institute of Organic Chemistry and Biochemistry, Czech Academy of Sciences, Flemingovo namesti 2, 166 10 Prague 6, Czech Republic; cihlova@uochb.cas.cz (B.C.); huskova@uochb.cas.cz (A.H.); boserle@uochb.cas.cz (J.B.); nencka@uochb.cas.cz (R.N.); boura@uochb.cas.cz (E.B.)

**Keywords:** high-throughput screening, virus, drug, discovery, papain-like, protease, SARS-CoV-2, flavivirus, TBEV

## Abstract

Spanish flu, polio epidemics, and the ongoing COVID-19 pandemic are the most profound examples of severe widespread diseases caused by RNA viruses. The coronavirus pandemic caused by severe acute respiratory syndrome coronavirus 2 (SARS-CoV-2) demands affordable and reliable assays for testing antivirals. To test inhibitors of viral proteases, we have developed an inexpensive high-throughput assay based on fluorescent energy transfer (FRET). We assayed an array of inhibitors for papain-like protease from SARS-CoV-2 and validated it on protease from the tick-borne encephalitis virus to emphasize its versatility. The reaction progress is monitored as loss of FRET signal of the substrate. This robust and reproducible assay can be used for testing the inhibitors in 96- or 384-well plates.

## 1. Introduction

RNA viruses are considered to be one of the most severe threats to the human population and quality of life [1]. Since the beginning of this millennium, we have witnessed at least 60 epidemic outbreaks around the world, mostly caused by RNA viruses. Excluding the ongoing COVID-19 pandemic, these have caused more than a million deaths. The viruses responsible include: influenza virus, Ebola virus (EBOV), Zika virus (ZIKV), yellow fever virus (YFV), dengue viruses (DENV), measles, and coronaviruses such as SARS, MERS, and SARS-CoV-2. The effect on human health is devastating, just as is the economic burden of these epidemics. The global cost of COVID-19 alone is astronomical and is predicted to surpass the GDPs of Germany, the UK, and France combined. Therefore, the global importance of targeting the RNA viruses is indisputable. 

The name and classification of RNA viruses originate from their genomic material which is composed of single-stranded or double-stranded RNA. Recent work has classified RNA viruses into five different orders with 47 families [2]. Although the nature of nucleic acid determines the classification of these viruses, more detailed classification is difficult due to the high mutation rate and recombination of the RNA viruses [3,4]. RNA viruses have the highest mutation rates among all viruses, which often leads to the development of resistance against antivirals. These high mutation rates generate new species, and two to three novel viruses are discovered every year [5]. Understanding how mutation rates drive and shape the evolution of new and potentially deadly viruses that can cross interspecies boundaries is an ongoing topic of scientific interest [6]. 

Upon the infection of the host cell, viral RNA is replicated by viral RNA-dependent RNA polymerase and translated into one or more polyproteins. Subsequently, these polyproteins are processed by viral and host proteases into individual structural and nonstructural proteins. The presence and the functionality of viral enzymes, particularly proteases and polymerases, are a vital step in the replication and spread of the virus. A potential antiviral drug is correspondingly aimed at inhibiting the viral enzymes, thus limiting the spread of the virus. However, approved antivirals against RNA viruses are generally lacking, and only a few RNA viruses can be currently treated by approved antivirals: the influenza virus, the respiratory syncytial virus, and hemorrhagic viruses such as the Lassa mammarenavirus or the hepatitis C virus (HCV). Without antiviral drugs, the only current form of treatment is supportive care, i.e., relieving pain and other symptoms. What has been raising hopes for developing RNA antivirals is the success of antiviral therapy against HCV [7]. The potent effects of these antivirals are mainly based on targeting proteases and polymerases, two of the essential viral enzymes. Basic research of these enzymes has formed the groundwork of rational drug design, enabling the development of specific molecules that bind and inhibit the enzymes. However, the action of novel synthetic molecules within the human body is rather unpredictable, leading to a high failure rate in the later stages of clinical trials. Additionally, the rapid rates of mutation further increase the overall fitness of the virus, making it resistant to antiviral agents, especially in the case of single-compound regimens. Greater antiviral effect is achieved by the combination of several antivirals, which usually include both polymerase and protease inhibitors.

In this study, we focused on two viral proteases of different catalytical types. Firstly, we employed the papain-like protease (PL^pro^) from SARS-CoV-2. Secondly, we used the nonstructural protein 3 protease (NS3^pro^), which is classified as a chymotrypsin-like serine protease, from the tick-borne encephalitis virus.

The papain-like protease (PL^pro^) is one of two proteases encoded by the coronavirus SARS-CoV-2, the cause of the current COVID-19 pandemic. PL^pro^ is a deubiquitinating-like (DUB-like) enzyme that negates the host interferon-induced cellular response by cleaving the interferon-stimulated gene 15 (ISG15). ISG15 is a small di-ubiquitin-like protein that is overexpressed during viral infection and covalently attached to newly synthesized proteins to mark the viral invader. ISG15 impedes the processes of the viral replication cycle. It is thought to block the formation of new viral particles due to the steric hindrance of ISG15 molecules attached to structural proteins that form the virion [8,9]. Therefore, cessation of the viral defence mediated by PL^pro^ makes it a bona fide therapeutic target. Moreover, the inhibition of the PL^pro^ enzyme from other coronaviruses has been demonstrated to suppress viral replication [10,11,12]. 

The nonstructural protein 3 protease (NS3^pro^) is the one protease encoded by the tick-borne encephalitis virus (TBEV). Tick-borne encephalitis (TBE) is the most significant flaviviral tick-borne disease that causes brain damage, paralysis, and even death. There are 10,000 to 12,000 reported cases of TBE each year but the worldwide risk of incidence is predicted to increase on account of the expansion of the tick population due to global warming and human mobility [13,14,15]. Other severe human pathogens in the Flavivirus genus include ZIKV, DENV, YFV, the West Nile virus (WNV) and the Japanese encephalitis virus (JEV). The virology and enzymology of flaviviruses have been studied extensively for the past 30 years. Thus, several kinetic and structural studies of their enzymes are available [16,17,18,19,20,21]. Significant structural and functional similarities between proteins of a single genus make these findings transferable and instrumental in other studies [22]. 

The flaviviral polyprotein is processed into three structural proteins (which form the envelope, membrane, and capsid), and seven nonstructural (NS) proteins (named NS1, NS2A, NS2B, NS3, NS4A, NS4B, and NS5). NS3 possesses two distinct activities, helicase and protease activity, on the N-terminus and C-terminus, respectively. NS3 protease (NS3^pro^) is a chymotrypsin-like serine protease whose cleavage site is specified by the sequence XX↓Y where X is a positively charged residue and Y is a small residue such as serine or glycine [23]. NS2B anchors the NS3 to the endoplasmic reticulum by its termini, and is known to participate in the protease reaction. 

Here, we report a general FRET-based method for high-throughput quantitative screening (HTS) of potential inhibitors, and for testing other enzymatic properties of viral and nonviral proteases. In addition to validating potential rationally designed molecules, HTS enables screening of already developed and clinically approved molecules that can potentially serve other purposes. As HTS plays an indispensable role in antiviral drug development, the methods must be as economical and rapid as possible while granting high reproducibility and robustness. FRET-based fluorescent assays are known to be a very versatile and widely used tool in molecular biology [24]. FRET is a nonradiative transfer of energy from one fluorophore (donor) to another chromophore (acceptor). Essentially, FRET-based assays report on the distance between the donor and the acceptor. While the fluorophores are in close proximity, the donor is excited and transfers its energy to the acceptor, which then produces a fluorescence signal at its characteristic wavelength. Upon separation, i.e., increase in the distance between the fluorophores, FRET is abolished, and the acceptor ceases to produce the signal (Figure 1). Thanks to its dependence on inverse sixth-power distance, FRET is a remarkably sensitive tool for measuring the dissociation of molecules. FRET is also influenced by an overlap between the emission spectrum of the donor and absorption spectrum of the acceptor, quantum yield of the donor, orientation between fluorophore dipoles, and other physical factors of the environment [25].

For our assay, we have selected common, attainable, and stable fluorescent proteins. The FRET pair of our choice consisted of a fluorescence donor (eGFP) and a fluorescence acceptor (mCherry) [26]. The GFP and mCherry FRET pair has been successfully used for imaging of proteoyltic cleavage in living cells [27]. Further examples are FRET pairs composed of fluorescent proteins that have been used for monitoring degradation by a proteasome [28]. The fluorescent pair can be readily changed according to the needs of a particular experiment. As a proof of principle, we have selected two medicinally significant targets, the PL^pro^ from SARS-CoV-2 and the NS2B-NS3 protease from TBEV, and tested several small molecules to optimize and validate the assay for an array of molecules and also different classes of proteases. 

## 2. Results

### 2.1. Preparation of TBEV Chymotrypsin-Like Protease and SARS-CoV2 Papain-Like Protease

The proteolytic activity of the TBEV NS3 protease is dependent on association with the NS2B cofactor. We prepared the NS2B–NS3 protease construct composed of NS2B (residues 45–96 linked with residues 116 to 131), followed by residues 1–190 of NS3 linked via a glycine-rich linker (Appendix A). This construct is similar to the constructs previously used to express other active and soluble flaviviral proteases, such as DENV [29]. The recombinant NS2B–NS3 protease was transformed and expressed in *E. coli* and purified to homogeneity. The junction between NS2B and NS3 was cleaved by autoproteolytic activity during the purification (Appendix A) as has been observed for ZIKV protease [18]. The cleavage site corresponds to the enzyme’s specificity for two basic residues followed by a small residue (here Arg–Arg–Ser), which directly confirmed the in vitro enzymatic activity. PL^pro^ from SARS-CoV-2 was recently recognized to facilitate specific cleavage of the di-ubiquitin-like protein ISG15 [30]. We produced the PL^pro^ enzyme recombinantly in *E.coli* and purified it to homogeneity.

### 2.2. Fluorogenic Substrates for NS2B–NS3^pro^ and PL^pro^ and Optimization of Activity Assays

A sequence of the substrate for the proteolytic reaction of NS2B–NS3^pro^ was designed to have a natural cleavage site present between the NS2B and NS3 of TBEV, flanked by GFP and mCherry fluorescent proteins on the N- and C-terminus, respectively (Figure 1). The fluorogenic PL^pro^ substrate was generated similarly as for NS2B–NS3^pro^, but the substrate molecule, ISG15, was cloned between the genes for the identical FRET pair (mCherry and eGFP) (Figure 1). Both fluorogenic substrates were recombinantly expressed in *E. coli* and purified to homogeneity. 

The activities of both NS2B–NS3^pro^ and PL^pro^ were tested in vitro using small-batch reactions under physiological conditions. The results of these reactions and the subsequent activity of both proteases were validated with an SDS-PAGE-based assay (Figure 1, Appendix A). Then, the fluorescent properties of NS2B–NS3^pro^ and PL^pro^ substrates were tested on a plate reader with adjustable wavelength (Tecan). In the case of FRET-based kinetics, both the increase in the donor fluorescence or decrease in the acceptor fluorescence can be followed to determine the rate of the reactions. We measured both the excitation and the emission spectra of the substrate and of the final product. This allowed us to determine the spectral conditions where the change in fluorescence was greatest upon the addition of the enzyme (Appendix A). The optimal excitation wavelength was 488 nm and the optimal emission wavelength was 610 nm. These parameters were used for all the FRET measurements.

### 2.3. Quantitative FRET Assay for Testing Potential Inhibitors of Viral Proteases in a High-Throughput Format

With established reaction conditions for the assay, we proceeded to measure the optimal enzyme concentration to achieve steady-state kinetics in the case of PL^pro^, where the initial reaction rate is linear, and single turnover in the case of NS2B–NS3^pro^. The single-turnover approach, which is done at the opposite limit to the steady-state kinetics, allowed us to rapidly investigate the inhibition of a protease which is catalytically less active. Overall, both conditions enabled the linear fit of initial reaction rates. This additionally gave sufficient time for the measurement of the reactions in a high-throughput mode in the entire plate. We tested serial dilutions of enzymes with constant amounts of the substrate and achieved final optimum concentrations of enzymes and their substrates used for all assays in a 384-well plate format, i.e., 20 nM enzyme and 1 μM substrate for PL^pro^, and 2.5 μM enzyme and 0.25 μM substrate were used for NS2B and NS3^pro^, respectively (Figure 1, Figure 2 and Figure 3). The excess of NS2B–NS3^pro^ over its substrate was used to compensate for the low activity of this protease in vitro and, therefore, to enable rapid screening.

We selected several inhibitors of PL^pro^ and NS2B–NS3^pro^ to be tested in our FRET-based assay in the 384-well plate format. All inhibitors were dissolved close to their solubility limits to achieve the maximum range of concentrations in individual assays. Typically, we prepared inhibitors at 3 mM, 10 mM, or 50 mM concentrations and, where possible, serial dilutions of inhibitors in the reaction buffer were used. The time allowed for the proteolytic reaction was 40–60 min for PL^pro^, and 3.5 h for NS2B–NS3^pro^. This allowed rapid measurement of the IC50 values in a broad range of concentrations. The assay setup was optimized such that the linear phase of the reaction would be sufficiently long to allow the mixing and measuring of the entire 384-well plate.

### 2.4. The Statistics and the Quality of the Assay

The average Z’ factor, the measure of the quality of the assay, for this assay was determined to be 0.49 ± 0.09. The average signal-to-noise (STN) ratio for PL^pro^ STN = 153 ± 27. The ratio of fluorescence of substrate and product was 1.56. The average signal-to-noise ratio for NS2B–NS3^pro^ substrate was STN = 138 ± 42. Signal-to-background (STB) was 54.6 for the substrate and 33.4 for the product.

### 2.5. The Potency of TBEV NS2B–NS3^pro^ Inhibitors

First, we thoroughly tested our assay with NS2B–NS3^pro^ from TBEV. TBEV protease has not been characterized structurally. However, it is known that the flaviviral NS2B–NS3^pro^ is a serine protease that is highly conserved within the Flavivirus genus [22]. Therefore, we chose to test several commercially available inhibitors that are sufficiently soluble and their interaction with the ZIKV, DENV and WNV proteases has been well characterized (DTNB, aprotinin) [17,18,31]. Additionally, leupeptin was picked to represent a broad-spectrum protease inhibitor [32]. The measured IC50 values are listed in Appendix A. The dose–response curves from the FRET measurements on the Tecan microplate reader and illustrative SDS-PAGE gels are shown in Figure 2. The detailed principle of FRET-based assays and the further analysis of the SDS-PAGE gel-based assay is exemplified in Appendix A.

In the case of leupeptin, there was no inhibition observed, as confirmed by the SDS-PAGE analysis. Aprotinin (also known as bovine pancreatic inhibitor), a competitive serine protease inhibitor, inhibited NS2B–NS3^pro^ with an IC50 value of 1.8 ± 0.2 μM. DTNB inhibited the flaviviral NS2B–NS3^pro^ enzyme with an IC50 value of 303 ± 54 μM. The DTNB molecule inhibits enzymes by forming disulfide bonds with cysteine residues on their surface. Therefore, reducing agents needed to be omitted from the reaction buffer and the enzyme stock solution (containing β-ME) was desalted to avoid premature reduction of DTNB. As expected, desalting resulted in increased instability of the enzyme and a tendency to precipitate. Rapid manipulation at low temperature was thus essential prior to the measurement. 

### 2.6. Length of the Glycine-Rich Linker between NS2B and NS3 Has a Minor Influence on Substrate Conversion of NS2B–NS3^pro^

To demonstrate the influence of the length of a glycine-rich linker on cleavage efficiency, we prepared two other constructs of NS2B–NS3^pro^ with longer (G_4_SG_2_SGSGS_2_GSGSG_3_) and shorter (GSG_3_) linkers. The proteolytic activity was tested in vitro with serial dilutions of the enzymes and constant amounts of the substrate. The reaction progress was analyzed densitometrically from SDS-PAGE gels. Results from the graphical representation show comparable cleavage efficiency for NS2B–NS3^pro^ with the original, longer, and shorter linker. Apparently, the proteolytic activity is influenced to a small extent (Appendix A). 

#### Small Molecule Inhibitors of SARS-CoV-2 PL^pro^, Antabuse Inhibits PLpro within the Nanomolar Range

For further establishment and validity of the high-throughput character of our assay, we selected 2-mercaptopurine (2-MP), 6-mercaptopurine (6-MP), JB24, and Antabuse, and tested them with the PL^pro^ enzyme and its substrate. 

All the reactions were tested in the same aforementioned conditions. We synthesized a compound JB24 as previously described as compound number 24 [33]. This molecule inhibited PL^pro^ significantly with an IC50 of 1.48 ± 0.39 μM. This molecule was utilized during the optimization of the assay for PL^pro^. The results were reproducible as individual titrations were measured with different batches of defrosted and newly diluted enzyme and a substrate (Figure 3). 

Next, Antabuse, 2-MP, and 6-MP were tested in order to measure their inhibition properties on PL^pro^. The inhibition of these compounds is sensitive to reducing agents. Therefore, the reducing agent had to be omitted from the reactions and it had to be removed from the reaction by desalting. Antabuse inhibition was 80 ± 38 nM in the absence of DTT or β-ME. Interestingly, 2-MP but not 6-MP had an inhibitory effect on PL^pro^ (IC50 = 0.82 ± 0.6 μM) (Figure 3). 2-MP inhibition was also diminished by the presence of reducing agent in the reaction buffer. 

## 3. Discussion

The assay presented in this study offers a rapid, inexpensive, and sensitive high-throughput screening of protease inhibitors. We demonstrate its effectivity and robustness on two viral proteases which differ both in specificity and activity. Both proteases represent significant therapeutic targets. The protease from SARS-CoV-2, a papain-like protease (PL^pro^), is a relatively fast-acting enzyme which prevents cellular antiviral response by its DUB activity. The second protease, formed by the nonstructural proteins 2B and 3 (NS2B–NS3^pro^), from the tick-borne encephalitis virus (TBEV), processes newly synthesized TBEV polyprotein during viral infection and displays low catalytic activity in vitro. The proteolytic reactions performed in vitro are significantly slower than the proteolytical cleavage in infected cells, especially in the case of the TBEV protease. Nonetheless, the results are accurate and enable us to establish an IC_50_ value for each inhibitor tested, which is the main starting point for drug design.

PL^pro^ is a cysteine protease and it contains a sulfhydryl group in the active site [30]. The presence of a reducing agent significantly increases the IC_50_ values of Antabuse and of 2-MP. This is not unprecedented as Antabuse is a thiol-reactive compound and it forms a covalent bond with catalytic cysteines of the enzymes it inhibits [34]. It is conceivable that 2-MP thiol group can also form a disulfide bridge with catalytic cysteine of PL^pro^. Therefore, the inhibition by both of these compounds is strongly affected by the presence of reducing agents [35].

To validate the assay quantitatively, we obtained reproducible values of the half-maximal inhibitory concentration (IC50). Aprotinin (also known as the bovine pancreatic inhibitor) inhibited NS2B–NS3^pro^ with an IC50 of 1.8 ± 0.2 μM at an enzyme concentration of 2.5 μM. Notably, this protease is inhibited significantly less efficiently than other proteases of flaviviruses. For aprotinin, the WNF protease displayed an IC50 of 20 nM when the enzyme concentration was 10 nM [36], and the DENV protease displayed an IC50 of 65 nM when the enzyme concentration was 1 μM [37]. The assay presented here can reveal compounds inhibiting the viral proteases in vitro. Nevertheless, these must be tested in cell-based assays to validate their potency.

We have demonstrated the versatility of this assay and that the fluorogenic substrate can be prepared easily by recombinant expression in *E. coli*. Moreover, the recombinant character of the substrate broadens the area for the assessment to practically any protease from any source, ranging from viruses to humans. In the case of our assay, eGFP and mCherry have been used for several reasons. The genes of these molecules are readily available in most laboratories. The excitation maximum of eGFP is in the vicinity of one of the most commonly used lasers (argon laser, 488 nm), which is available on most instruments. Our assays with enzymes from very distinct kinetics have demonstrated the practicality of our choice of the FRET pair (eGFP and mCherry) in combination within the gel-based assays. Both eGFP and mCherry are very stable proteins with sufficiently good FRET efficiency for the assay. In the case of further revalidation in the PAGE gel, the larger spectral differences are advantageous. Importantly, for enhanced sensitivity of this type of assay, other FRET pairs such as the CyPet–YPet pair could be considered. This pair offers greater FRET emission gains, especially when instrumental setup allows for excitation at 414 nm and the collection of emissions at 530 nm [38]. Surprisingly, until the present day, FRET-based HTS protease assays overwhelmingly utilized purely synthetic FRET pairs with fluorescent moieties flanking the peptide sequence with a cleavage site (e.g., in a recent case for MERS PL^pro^ and SARS PL^pro^) [23]. Although this is a very elegant and often more efficient solution, the cost and versatility of recombinant FRET-based assays exceeds such an approach with large, more complex substrates and their mutations or varieties of different constructs can be easily prepared.

In comparison with radioisotope-based methods, the fluorescence-based methods are safe to use, easy to operate, and the measurements and possible evaluation are carried out in real time. In most cases, our fluorescent assay does not require further manipulation and analysis of the sample, e.g., separation of reactants from products and measurements. On the other hand, interference from other components of the assay may be an issue in the case of fluorescent assays and has to be taken into consideration. These components may interfere with the assay by altering the amount of fluorescent signal and decreasing the sensitivity of the assay. In the proper setup of our assay, the photobleaching of the components can be corrected for by running the controls for the assay, e.g., fluorescent substrate without the enzyme. It is noteworthy that the fluorescent substrate, cofactors, or inhibitors may interfere with the assay in such a way that FRET-based results are not suitable for evaluation. In such a case, assays can be resolved on a PAGE gel, with the cost of a lower throughput. 

The fluorescent assay using SDS-PAGE gel demonstrated in this work represents reaction progress at the last time point of the reaction where the reaction was stopped by the SDS sample buffer (Figure 2 and Figure 3, Appendix A). The amount of the substrate cleaved corresponds to the appearance of the product band, and the thinning of the substrate band becomes apparent only in a later stage of the reaction, therefore the densitometric analysis is necessary to evaluate these experiments. The analyzed gel-based data were fitted dose–response curves and estimated IC_50_ values corresponded relatively well to FRET measurements, confirming the validity of this FRET-based assay. These IC_50_ values are not to be directly compared with more precise FRET measurements determined from initial rates of the reaction. Such a comparison is only shown illustratively in Appendix A. All of the gel-based assays were only meant to validate the inhibition of our approach and FRET-based HTS assay. We have also demonstrated the utility of the SDS-PAGE gel-based fluorescent method as an alternative technique when FRET-based experiments cannot be used.

Other fluorescent techniques besides FRET-based methods may be considered to be applicable for measurements of the rate of proteolysis, especially in similar sample setups where the large fluorescent substrate is proteolytically cleaved into one or several small fluorescent molecules. In this case, fluorescence anisotropy or polarization (FA/FP), homogeneous time-resolved fluorescence (HTRF), and fluorescence correlation spectroscopy (FCS) are among these options. There are advantages and disadvantages to all of these methods. However, there is one key aspect that outweighs these excellent techniques in favor of FRET-based techniques and that is their suitability for the HTS. 

The recent advances in FCS allow measurements in high-throughput mode, but the instrument cost, complex evaluation, and speed in processing of the samples may be a major obstacle in the use of such a method [39]. On the other hand, when the number of available components is limited and there are low volumes of the sample, confocal microscopy may be favorable. Although FP and HTRF methods can also be used in high-throughput screening (HTS), they require more complex instrumentation and evaluation of the results. In this case, low cost and greater versatility, modality, and possibly greater sensitivity favor the FRET-based method [40].

The other fluorescent-based methods have plenty of advantages, but one of the overwhelming factors in favor of this assay is its low cost and versatility of preparation. This assay can be easily performed in a typical laboratory and only requires a fluorescent plate reader. The fluorescent parameters of the substrate may be tailored to suit the available instruments of a particular researcher and to the given application, e.g., mutagenic study of the variability of the substrate or testing the enzyme kinetics. Since the substrate is recombinant, it avoids costly synthesis of specific substrates with synthetic fluorescent probes that might be essential for the aforementioned methods. Fluorescence assays typically offer the best balance between cost and sensitivity in high-throughput screening (HTS) experiments.

Rapid response to threats posed by viral pathogens is and will clearly continue to be one of the most important challenges for our globalized society. This response is associated with our ability to quickly design and prepare new antivirals that will be able to affect these pathogens at their weakest points. Experience with the discovery of drugs against both HIV and HCV illustrates that viral proteases are important drug targets. Here, we have shown that it is possible to quickly and efficiently develop an assay against various viral proteases and convert it to the HTS format. Therefore, the assay developed here is a very useful tool for early drug discovery and can be quickly designed and used for both drug repurposing and the identification of completely new protease inhibitors in the fight against various viral pathogens.

## 4. Methods

### 4.1. Cloning, Expression, and Purification of the Recombinant NS2B–NS3^pro^ and PL^pro^


The DNA sequence encoding the TBEV NS2B–NS3^pro^ enzyme (strain Hypr; GeneBank: KP716978.1) was commercially synthesized (Invitrogen) and encoded NS2B (residues 45–96), a GGGGSGGGG linker, and NS3 (residues 116-131) followed by a 6xHis-tag. The NS2B–NS3pro encoding gene was cloned into the NcoI and the NotI sites of a pRSFD vector (Novagen). The vector was transformed into *Escherichia coli* (*E. coli*) NiCo21 (DE3) cells and cultured in LB medium with 40 μg/mL of kanamycin. The culture was left to shake overnight at 37 °C and then used to inoculate ZY5052 autoinduction media. After reaching an optical density of 0.6–0.8 (OD600) at 37 °C, the temperature was lowered to 18 °C and the culture was grown overnight. The cells were lysed by sonication in a lysis buffer containing 20 mM Tris-HCl pH = 8; 300 mM NaCl; 20 mM imidazole; 10% glycerol; and 3 mM β-mercaptoethanol (β-ME). The supernatant was separated by centrifugation, incubated with 2 mL of Ni-NTA resin (Machery-Nagel) and extensively washed with the lysis buffer using the batch technique. The slurry was loaded on the column and the protein was eluted with the lysis buffer supplemented with 300 mM imidazole pH = 8.0. The eluate was further purified using the size-exclusion Superdex 75 HiLoad 16/600 column (GE Healthcare) with a gel filtration buffer (20 mM Tris-HCl pH = 8; 300 mM NaCl; 10% glycerol; and 3 mM β-ME). Protein was desalted on a HiPrep 26/10 desalting column (GE Healthcare) and loaded on an anion exchange HiTrap Q HP column. The protein was eluted by a salt gradient in buffer A (20 mM Tris-HCl pH = 8, 50 mM NaCl, 10% glycerol, and 3 mM β-ME). The purity of the protein was verified on SDS-PAGE in 15% acrylamide:bis-acrylamide gel, stained with Coomassie brilliant blue (Appendix A). The protein was concentrated to 2.8 mg/mL frozen in N2(l) and kept at −80 °C. 

The gene encoding the PL^pro^ (also known as nsp3) protein from SARS-CoV-2 (YP_009725299.1) was also synthesized commercially (Invitrogen) and cloned into the pSUMO1 vector with N-terminal 8xHis conjugated with yeast SUMO, forming a fusion solubility/affinity tag. The plasmid was transformed into *E. coli* NiCo21 (DE3) and expressed in ZY5052 autoinduction media supplemented with 50 μM ZnSO4 and affinity-purified identically to the NS2B–NS3^pro^. The protein was desalted on a HiPrep 26/10 desalting column (GE Healthcare) and loaded on an anion exchange HiTrap Q HP column (GE Healthcare). Next, the 8xHis-SUMO-tag was cleaved using SUMO protease from yeast (Ulp1), and after overnight incubation at 4 °C, the sample was loaded onto a HisTrap HP, equilibrated in a lysis buffer. Unbound fractions containing PL^pro^ were pooled, concentrated, and loaded on a Superdex 75 HiLoad 16/600 column (GE Healthcare) equilibrated with 20 mM Tris-HCl pH = 7.4; 50 mM NaCl; 10% glycerol; and 3 mM β-ME. The protein was checked on 15% SDS PAGE gel (Appendix A), concentrated, and frozen in N2(l) and kept at −80 °C. 

### 4.2. Preparation of Fluorescent Substrates eGFP-RSSRRSDLVFS-mCherry and mCherry-ISG15-eGFP 

A substrate for the proteolytic reaction of NS2B–NS3^pro^ was designed to have a sequence (RSSRRSDLVFS) derived from a cleavage site present between the NS2B and NS3 of TBEV, flanked by GFP and mCherry fluorophores, resulting in a plasmid encoding for GFP-RSSRRSDLVFS-mCherry. The plasmid was prepared by restriction cloning. First, a gene encoding for GFP was cloned in the pHis2 vector. In the second step, the NS2B–NS3^pro^ site and mCherry were added. The plasmid was transformed into the *E. coli* NiCo21 (DE3), and the protein was expressed in LB medium supplemented with 0.1 μg/mL of ampicillin, 10 μM ZnSO4, 1 mM of MgCl2 and MgSO4, 0.25 mM KCl, and 15 μM FeCl2 dissolved in citric acid. The medium was incubated in a shaker at 37 °C. After reaching an OD600 of 0.4, the temperature was lowered to 25 °C, at which point 0.3 μM IPTG was added to initiate the expression. The temperature was immediately lowered to 18 °C, and the culture was grown overnight. The purification steps were similar to those used to produce the NS2B–NS3^pro^. The cells were briefly lysed by sonication in a lysis buffer (20 mM Tris-HCl pH = 8; 300 mM NaCl; 20 mM imidazole; 10% (*v*/*v*) glycerol; and 3 mM β-ME), and then supplemented with one tablet/L of the complete mini EDTA-free protease inhibitor cocktail. The supernatant was purified in a Ni-NTA column, Superdex 75 HiLoad 16/600 column, HiPrep 26/10 desalting column, and HiTrap Q HP column, respectively, using the same buffers and procedures as described above. Additionally, NaCl was added to a final concentration of 800 mM and the protein was further purified on the Superdex 75 HiLoad 16/600 column. The enzyme was stored at −80 °C.

Human ISG15 was the substrate of SARS-CoV-2 PL^pro^. The gene encoding ISG15 was subcloned in between genes encoding the FRET pair consisting of mCherry and eGFP (Figure 1a). All these components were amplified using PCR, and they were cloned into the plasmid pET-24a using a Gibson assembly [41]. The plasmid was transformed into *E. coli* NiCo21 (DE3), expressed, and purified to homogeneity. The final recombinant protein used for the assays contained an N-terminal 6 x His-Tag, mCherry, a cleavage site for TEV protease, ISG15, and eGFP (mCherry-ISG15-eGFP). The TEV site was included for validation and versatility of the substrate.

### 4.3. FRET-Based Assays of the NS2B–NS3^pro^ and PL^pro^ Activity and Inhibition

The reactions of the proteases and their FRET substrates were performed in 80 μL in black 384-well plates. In the case of NS2B–NS3^pro^ reactions, there was a 2.5 μM protease and 0.25 μM substrate. In the case of the PL^pro^ reaction, there was 20 nM protease and 1 μM substrate. The reaction buffer contained 20 mM Tris-HCl pH = 7, 10 mM NaCl, 3 mM β-ME for NS2B-NS3^pro^ or 20 mM Tris-HCl pH = 7.4, 50 mM NaCl, 10 % glycerol, and 3 mM β-ME for PL^pro^. The reaction conditions were optimized to fulfill the HTS character. In the case of the inhibitor 5,5-dithio-bis-(2-nitrobenzoic acid) (DTNB), β-ME was omitted from the buffer due to the easy reduction of DTNB to 2-nitro-5-thiobenzoate (TNB). Similarly, β-ME was omitted in the cases of 2-MP, 6-MP, and Antabuse. The enzyme stock solution was desalted using the MicroSpin G-25 columns (Cytiva) when testing the easily reduced inhibitors. The titration series of the inhibitor was performed in a mixture containing a constant concentration of the substrate. Reactions were initiated by mixing equal volumes (i.e., 40 μL) of the mixture containing the substrate and inhibitor with a mixture containing the enzyme. 

The 384-well plate layout of the reaction allowed for the following setup: two negative control reactions, twelve to sixteen different reactions with a gradient of concentration, and one positive control with both enzyme and substrate. In this set, the reactions were measured in three to four technical replicates, and all of the sets of these reactions were repeated at least three times. Negative control only contained the FRET substrate, and otherwise were treated identically to other reactions. The data from negative control were used to subtract the fluorescence changes from the other datasets in order to compensate for other processes e.g., photobleaching. The positive control contained both the FRET substrate and the enzyme, without an inhibitor. This control was used to determine the maximum activity of the enzyme in the particular reaction set. The data were normalized to express the percentage of inhibition according to the difference between positive and negative control. Each measurement was carried out in technical triplicates. Each measurement was replicated at least three times. Measurements were performed using the Tecan microplate reader at 25 °C for 3.5 h (NS2B–NS3^pro^) or 40–60 min (PL^pro^). To prevent the NS2B–NS3^pro^ reaction mixture from evaporating, the plates were covered with a transparent Crystal Clear Sealing Tape (Hampton research).

In the reaction, the decreasing fluorescence (FRET) intensity of mCherry was monitored at an emission wavelength of 610 nm. The time interval between measurements was 30 min in the case of NS2B–NS3^pro^ and 30 s to 60 s in the case of PL^pro^. The excitation wavelength was 488 nm, with a 5 nm bandwidth for both slits. The lag time was zero, the integration time was 20 μs, and the settle time was 10 ms for 400 Hz of flash frequency. An optimal 96% gain was calculated in every measurement. A Z-position was set at a 20,000 μm height. For each time point, 10 flashes were integrated and combined with the multiple reads per well with 100 μm offset from the border of the well. Immediately after the measurement, 20 μL of 5× SDS sample buffer (60 mM Tris pH 6.8, 25% (*v*/*v*) glycerol, 2.9% (*w*/*w*) SDS, 0.1% (*v*/*v*) Bromphenol Blue, 714 mM β-ME l) was added to terminate the reaction and validate the progress on the 15% reducing SDS-PAGE.

### 4.4. Analysis of Reaction Progress by Gel-Based Assay

The SDS-PAGE gels with resolved reaction mixtures were scanned on the Typhoon Biomolecular Imager (GE Healthcare), with a green laser (532 nm) and long pass red filter (LPR 660 nm) were used. The images were quantified using ImageQuant TL. Lines were selected manually and the background subtraction was performed using the rolling ball method. Bands with constant dimensions encompassing substrate and product were selected manually. The resulting substrate conversion was used in a similar manner as for fluorescent assays. The percentage of the inhibition of individual reactions was calculated relative to the control without the inhibitor. The exemplary evaluation of several gels is demonstrated in Appendix A.

### 4.5. Determination of Z’ Factor and Signal-to-Noise Ratio

The Z’ factor is a measure of the quantity of the assay for HTS. To determine the Z’ factor, 24 replicates of positive control and negative control were measured under the same conditions as in the inhibition measurements. Therefore, the reactions were performed in 80 μL in black 384-well plates, at 25 °C, and for the appropriate time. Measurements with NS2B–NS3^pro^ took 3.5 h and 60 min for PL^pro^. In the positive control, there was 2.5 μM protease and 0.25 μM substrate in the case of NS2B–NS3^pro^, and 20 nM protease and 1 μM substrate in the case of the PL^pro^. The negative control contained the same amount of substrate and no enzyme. The measurements were performed three times. The Z’ factor was calculated according to the method described which defined the Z’ factor as:(1)$$\;Z\prime \mathrm{factor}=1–\frac{3\sigma_{P}-3\sigma_{N}}{\left|\mu_{P}–\mu_{N}\right|}$$
where *σ_P_* and *σ_N_* are the standard deviations of positive and negative controls, respectively, and *μ_P_* and *μ_N_* are the means of positive and negative controls, respectively [42]. Signal-to-noise ratio (S/N or STN) was calculated by the ratio of the average signal over the standard deviation of the measured signal, and the average STN was calculated from three different and independent measurements. Signal to background (S/N or STB) was calculated from the average signal at 610 nm to the average background at 750 nm. STB and values were determined for the substrate and the product (Appendix A).

### 4.6. Determination of the Inhibitor Potency

To quantify the inhibitory effect of the tested molecules, we determined the half-maximal inhibitory concentration (IC50) from the measured data. The rate of the reaction was estimated from the slope of the initial part of the reaction as the decline of relative emission intensity in time (3.5 h for NS2B–NS3^pro^ measured in 30 min intervals) (Figure 1, Appendix A). For PL^pro^, 40–60 min reactions were measured in 30–60 s intervals and the slope was estimated from the initial 15 min. The average slope of the negative control (no enzyme added to the substrate reaction mix) that represented the FRET signal alone was subtracted from each slope of inhibition reactions and the positive control (Appendix A). The rate of the positive control corresponded to maximal enzymatic activity. The relative percentage of the inhibition was calculated using corrected positive controls (Appendix A). All normalized data were plotted and fitted against the log of the concentration of the inhibitor to give an IC_50_ curve of the specific inhibitor.

### 4.7. Influence of Length of the Glycine-Rich Linker between NS2B and NS3

Two additional NS2B–NS3^pro^ constructs with different lengths of the peptide linker between NS2B and NS3 chains were prepared (longer linker, G_4_SG_2_SGSGS_2_GSGSG_3_ and shorter linker, G_1_SG_3_). These constructs were used to perform reactions with NS2B–NS3 FRET substrates as described above. Briefly, the titration series contained 0.063 to 4 μM enzyme and constant concentration of the substrate, 0.25 μM. The reaction buffer contained 20 mM Tris-HCl pH = 7, 10 mM NaCl, 3 mM β-ME. Reactions were performed in triplicate at 25 °C in 80 μL in the dark. After 6 h, reactions were terminated with 5× SDS sample buffer and the reactions resolved on SDS-PAGE. The gel was scanned, using a laser at 532 nm and filter at 660 nm on the Amersham Typhoon Biomolecular Imager (GE Healthcare). The gels were analyzed using the commercial ImageQuant TL software as described above in the section on SDS-PAGE analysis.

## Figures and Tables

**Figure 1 molecules-26-03792-f001:**
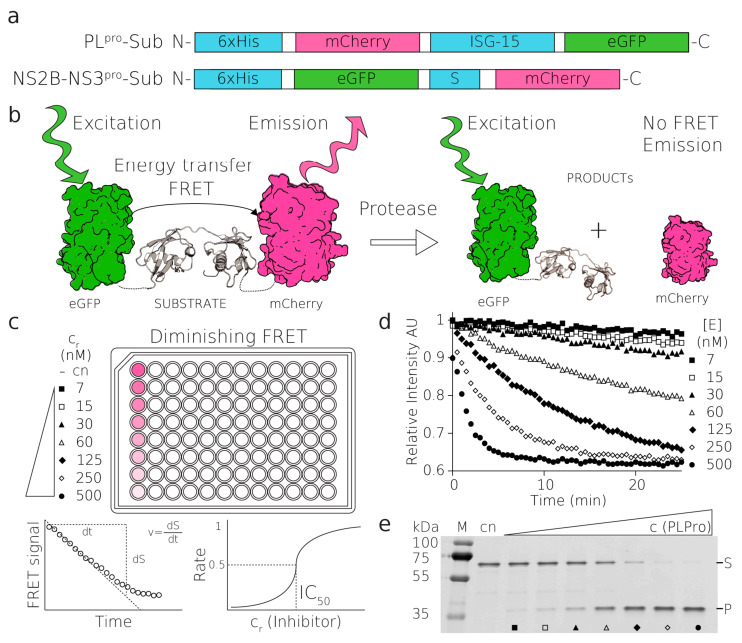
Substrates, principles of FRET assay. (**a**) The fluorescent PL^pro^ and NS2B-NS3^pro^ substrates, (**b**) schematics of the FRET-based proteolytic assay, eGFP is excited with a green light at 488 nm, FRET transfers the excitation to mCherry which emits the light that is detected. After proteolytic cleavage, this FRET signal is abolished, (**c**,**d**) the reaction of PL^pro^ substrate with a serial dilution of PL^pro^ enzyme (500–7 nM), (**c**) schematics of reactions followed in the fluorescent plate reader, (**d**) these reactions were loaded on SDS-PAGE gel and visualized on a fluorescent scanner (**e**), where (S) and (P) donate substrate and product bands and full-length gel are shown in Appendix A.

**Figure 2 molecules-26-03792-f002:**
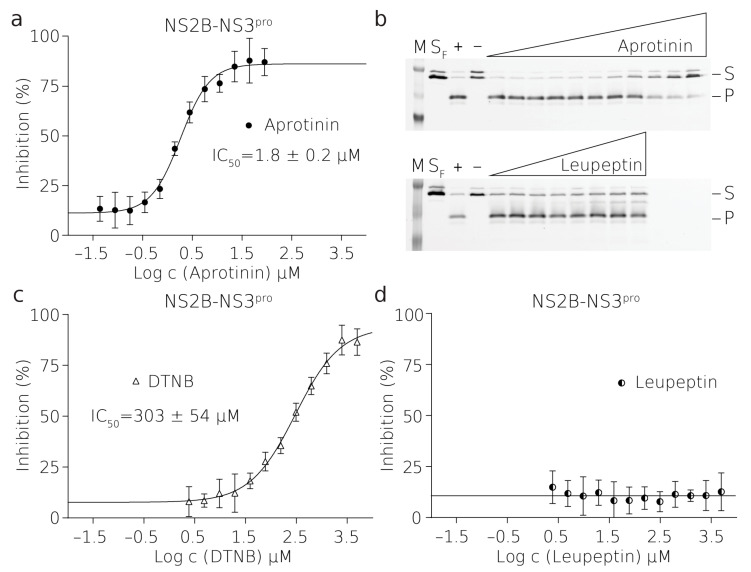
Inhibition of TBEV NS2B–NS3^pro^ by small molecules. Curves derived from the FRET inhibition assays of (**a**) aprotinin, (**c**) DTNB, and (**d**) leupeptin. Each curve is derived from three experiments. (**b**) Validation by SDS-PAGE analysis showing the successful inhibition by aprotinin (top) and no inhibition by leupeptin (bellow) where (M) is a marker, (SF) is fresh substrate at reaction concentration, (+) is positive control reaction where there was no inhibitor present, (–) is negative control reaction where there was no enzyme present, (S) is the substrate, (P) is the product. The illustrative evaluations of the validation SDS-PAGE gels and more details are shown in Appendix A.

**Figure 3 molecules-26-03792-f003:**
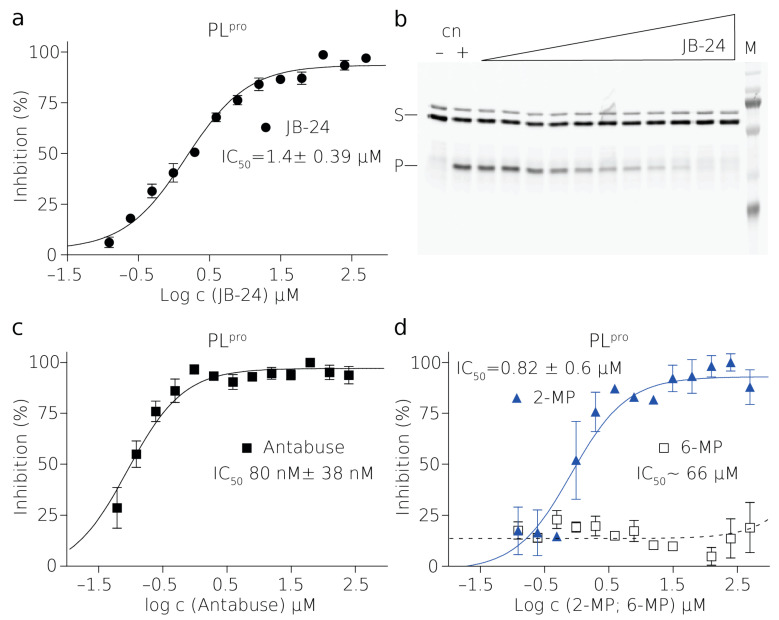
Inhibition of SARS-CoV2 PL^pro^ by small molecules. Curves derived from the FRET inhibition assays of (**a**) JB24, (**c**) Antabuse, and (**d**) 2-MP and 6-MP. Each curve is derived from three experiments. (**b**) SDS-PAGE gel from one of the titrations with JB24 where (M) is a marker, (+) is a positive control reaction in which there was no inhibitor present, (–) is a negative control reaction where there was no enzyme present, (S) is the substrate, and (P) is the product. The illustrative evaluations of the SDS-PAGE gels resolving the reaction progress and more details are shown in Appendix A.

## Data Availability

All data are contained within the manuscript and Appendix A.

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
