# Peer review of "High-Throughput Fluorescent Assay for Inhibitor Screening of Proteases from RNA Viruses"

_molecules, 2021, doi:10.3390/molecules26133792_

Round 1

Reviewer 1 Report

Currently, RNA viruses have seriously threatened human health, especially the current SRAS-CoV-2 pandemic. Screening of important RNA virus chemical drugs is extremely important for its prevention, control and treatment. Based on the fluorescent energy transfer (FRET), the author used SARS-CoV- 2 papain-like protease (PLpro) and Tick-borne encephalitis virus NS2B-NS3pro proteases as research models to develop and optimize a high-throughput screening system for screening small molecule compounds that inhibit the above-mentioned proteases. Overall, this work has important scientific significance for the development of drugs against the infection of RNA viruses.

Author Response

Response to Reviewer 1 Comments

Currently, RNA viruses have seriously threatened human health, especially the current SRAS-CoV-2 pandemic. Screening of important RNA virus chemical drugs is extremely important for its prevention, control and treatment. Based on the fluorescent energy transfer (FRET), the author used SARS-CoV- 2 papain-like protease (PLpro) and Tick-borne encephalitis virus NS2B-NS3pro proteases as research models to develop and optimize a high-throughput screening system for screening small molecule compounds that inhibit the above-mentioned proteases. Overall, this work has important scientific significance for the development of drugs against the infection of RNA viruses.

Response:

We thank the reviewer 1 for a review of our manuscript. We have included new text and some references to satisfy all of the reviewers comments. As a result the references may not be in order. We have agreed with the editor to label them accordingly and the editorial office would be renumbering them later on. The new references are cited by name and the year and in the brackets e.g. [Lipsky 2001]. The list of references is added as at the very end of the manuscript.

Reviewer 2 Report

Cihlova  and the co-authors established the reporter system suitable for FRET read-out and compatible with identification of inhibitors of viral proteases. Specifically, two reporters for papain-like protease of SARS-CoV-2 and protease of tick-borne encephalitis virus (TBEV) were described. In the light of the current global pandemics, as well as other RNA-virus induced infections, this work is truly interesting and necessary.

The first part of the introduction concerned with RNA viruses in general is very well written. The second part dealing with the chosen viruses and their proteases is rather scattered and needs to be consolidated. In addition, the following issues need to be addresses in order to get it published:

  1. It is better to move the description of the Figure 1 to the Results
  2. The comparison to already available FRET reporters be it specifically for proteases or that of other enzymes needs to be done in the Introduction and, if possible, in the Discussion
  3. References to the Supplemental Figures are inaccurate throughout the whole text. Please, check and re-order
  4. The effect of the reducing agents and IC50 values of the inhibitors is presented as a sided issue. It is important to discuss your observations in more details
  5. The applicability to high-throughput is not clear when 3 compounds are tested in 384 well format. Please, discuss how the assay can be scaled-up to be suitable for the high-throughput applications
  6. It is not clear whether the references 33 and 34 are discussing the same compounds as in your study or they refer to the same enzymes (lines 277-280). Please, clarify
  7. It is also not clear from the Discussion why FRET is more powerful in comparison to other fluorescent-based methods. The table with Pros and Cons of the methods in the Discussion could be a good option

Author Response

Response to Reviewer 2 Comments

The first part of the introduction concerned with RNA viruses in general is very well written. The second part dealing with the chosen viruses and their proteases is rather scattered and needs to be consolidated. In addition, the following issues need to be addresses in order to get it published:

Responses:

We thank the reviewer 2 for detailed and valid comments; we agree that the introduction and result sections needed consolidation. The second section of introduction was consolidated to improve the flow of the text. Since we have included new text and some references to satisfy all of the reviewers as a result the references may not be in order. We have agreed with the editor to label them accordingly and the editorial office would be renumbering them later on. The new references are cited by name and the year and in the brackets e.g. [Lipsky 2001]. The list of “new references” is added as at the very end of the manuscript.

  1. It is better to move the description of the Figure 1 to the Results

We kindly ask the editorial office to move the entire Figure 1 with the description (legend) to the next page.

  1. The comparison to already available FRET reporters be it specifically for proteases or that of other enzymes needs to be done in the Introduction and, if possible, in the Discussion

We have changed appropriate section of the introduction and discussion.

Introduction:

“…FRET-based fluorescent assays are known

to be a very versatile and widely used tool in molecular biology [Carnero2006]. …

…For our assay, we have selected common, attainable and stable fluorescent proteins. The FRET pair of our choice consisted of a fluorescence donor (eGFP) and a fluorescence acceptor (mCherry) [Albertazi2009]. GFP and mCherry FRET pair has been successfully used for imaging of proteoyltic cleavage in living cells [Jin2011]. Further examples were FRET pairs composed of fluorescent proteins have been used for monitoring degradation by proteasome [Neefjes2004].” 

Discussion:

Surprisingly until present day overwhelmingly FRET-based HTS protease assays utilize purely synthetic FRET pairs with fluorescent moieties flanking peptide sequence with cleavage site (e.g. in a recent case for MERS PLpro and SARS PLpro) [23]. Although this is a very elegant and often more efficient solution, the cost and versatility of recombinant FRET-based assay exceeds such an approach with large more complex substrates and their mutations or varieties of different constructs can be easily prepared..“

  1. References to the Supplemental Figures are inaccurate throughout the whole text. Please, check and re-order

We apologise for this error, all of references for the supplementary figures 1 and two were annotated by number 1. We have gone throughout a text several times and hopefully corrected all of these annotations.

  1. The effect of the reducing agents and IC50 values of the inhibitors is presented as a sided issue. It is important to discuss your observations in more details

The effect of reducing agents is interesting, although in our opinion it is outside of the scope of this manuscript as this work is not intended to study the mechanisms of particular inhibitors. However, we have included a paragraph in discussion devoted to this topic.

“Since PLpro is a cysteine protease, it contains sulfhydryl group in the active site [28]. The presence of a reducing agent significantly increases the IC50 values of Antabuse and of 2-MP. This is not unprecedented as Antabuse is a thiol-reactive compound and it forms covalent bond with catalytic cysteines of the enzymes it inhibits [Galkin2014, Lipsky2001]. It is conceivable that 2-MP thiol group can also form a disulfide bridge with catalytic cysteine of PLpro. Therefore, the inhibition by both of these compounds is strongly affected by the presence of reducing agents [32].”

  1. The applicability to high-throughput is not clear when 3 compounds are tested in 384 well format. Please, discuss how the assay can be scaled-up to be suitable for the high-throughput applications

We have realized that the way the paragraph was constructed was confusing. The entire assay has been optimised in such a way that a 384-well plate can be fully used to measure in every well. Moreover, in this particular experiment we have measured only 3 inhibitors. Therefore, we have removed the following sentence: “Up to 3 different inhibitors could be measured simultaneously” to mend this error.

Note:“…This allowed rapid measurement of the IC50 values in a broad range of concentrations. The assay setup was optimized such that the linear phase of the reaction would be sufficiently long to allow the mixing and measuring of the entire 384-well plate.”

  1. It is not clear whether the references 33 and 34 are discussing the same compounds as in your study or they refer to the same enzymes (lines 277-280). Please, clarify

This is the case of the same compound, aprotinin→ this has been mentioned and therefore clarified in text.

  1. It is also not clear from the Discussion why FRET is more powerful in comparison to other fluorescent-based methods. The table with Pros and Cons of the methods in the Discussion could be a good option

We thank the reviewer for an interesting point. We also appreciate that advantages and disadvantages of this assay in comparison with other techniques are important. We devoted an entire section of discussion to this topic. We believe that clearly stating the advantages of FRET techniques is sufficient.

New References         

           Albertazzi, L.; Arosio, D.; Marchetti, L.; Ricci, F.; Beltram, F. Quantitative FRET analysis with the E0GFP-mCherry fluorescent protein pair. Photochem. Photobiol. 2009, 85, 287–297, doi:10.1111/j.1751-1097.2008.00435.x.

Carnero, A. High throughput screening in drug discovery. Clin. Transl. Oncol. 2006, 8, 482–490, doi:10.1007/s12094-006-0048-2.

           Galkin, A.; Kulakova, L.; Lim, K.; Chen, C.Z.; Zheng, W.; Turko, I. V.; Herzberg, O. Structural basis for inactivation of Giardia lamblia carbamate Kinase by disulfiram. J. Biol. Chem. 2014, 289, 10502–10509, doi:10.1074/jbc.M114.553123.

           Jin, S.; Ellis, E.; Veetil, J. V.; Yao, H.; Ye, K. Visualization of human immunodeficiency virus protease inhibition using a novel Förster resonance energy transfer molecular probe. Biotechnol. Prog. 2011, 27, 1107–1114, doi:10.1002/btpr.628.

           Neefjes, J.; Dantuma, N.P. Fluorescent probes for proteolysis: Tools for drug discovery. Nat. Rev. Drug Discov. 2004, 3, 58–69, doi:10.1038/nrd1282.

Reviewer 3 Report

The manuscript by Cihlova and Huskova et al. describes a cost effective method to screen for inhibitors of viral proteases, which holds potential for new antivirals discovery in preparation for the on-going and future pandemics. I don’t have major concerns, except that the result section is too long without subsections. It would be better if the section can be re-organized so that it is easier and clearer to read. There are also too many method details in the result section, which should be moved to the method section.

Author Response

Response to Reviewer 3 Comments

The manuscript by Cihlova and Huskova et al. describes a cost effective method to screen for inhibitors of viral proteases, which holds potential for new antivirals discovery in preparation for the on-going and future pandemics. I don’t have major concerns, except that the result section is too long without subsections. It would be better if the section can be re-organized so that it is easier and clearer to read. There are also too many method details in the result section, which should be moved to the method section.  

Responses:

We thank the reviewer 3 for his review and comments; we agree that result section may be too wordy and full of experimental details. The result section was truncated and reorganised, overly detailed methodology in result sections was merged with method section.

We subdevided the entire result section into subsections with brief titles to improve the clarity and navigation of the reader thoughtout the manuscript.

Additionally, we have included new text and some references to satisfy all of the reviewers comments and as a result the references may not be in order. We have agreed with the editor to label them accordingly and the editorial office would be renumbering them later on. The new references are cited by name and the year in the brackets e.g. [Lipsky 2001]. The list of “new references” is added as at the very end of the manuscript.